# Student eXperience: A Systematic Literature Review

**Nicolás Matus** * [ID], **Cristian Rusu** * [ID] and **Sandra Cano**

School of Computer Engineering, Pontificia Universidad Católica de Valparaíso, Valparaíso 2340000, Chile; sandra.cano@pucv.cl
* Correspondence: nicolas.matus.p@mail.pucv.cl (N.M.); cristian.rusu@pucv.cl (C.R.)

**Abstract:** Students' experiences have been covered by a large number of studies in different areas. Even so, the concept of student experience (SX) is diffuse, as it does not have a widely accepted meaning and is often shaped to the specific purposes of each study. Understanding this concept allows educational institutions to better address the needs of students. For this reason, we conducted a systematic literature review addressing the concept of SX in higher education, specifically aiming at undergraduate students. In this work, we approach the concept of SX from the perspective of customer experience (CX), based on the premise that students are users of higher education institutions' products, systems and/or services. We reviewed articles published between 2011 and 2021, indexed in five databases (Scopus, Web of Sciences, ACM digital, IEEE Xplore and Science Direct), trying to address research questions concerning: (1) the SX definition; (2) dimensions, attributes and factors that influence SX; and (3) methods used to evaluate the SX. We selected 65 articles and analyzed various SX definitions, as well as scales and surveys to evaluate SX, mainly relating to satisfaction and quality in higher education. We propose a holistic definition of SX and recommend ways to achieve its better analysis.

**Keywords:** student experience; customer experience; higher education

## 1. Introduction

The satisfaction of people's needs is undoubtedly an element of vital importance for the consolidation of a brand or company in the market. This may become even more complex when considering the perceptions of customers that interact with an organization. Customer experience (CX) refers to people's expectations, emotions and interactions with a brand through systems, products, and services, and should be a priority in the market; it has been pointed out that its proper administration can lead to a differential advantage for service organizations [1].

CX requires observing the interaction between a customer and a brand, which is done through the contact points between them. Touchpoints are the instances in which the customer comes into contact with an organization throughout the whole customer "journey". These touchpoints can be both physical and logical. Touchpoints consider communication devices and channels, as well as tasks performed by customers [2].

When we think that students consume products and services, in addition to interacting with university software systems, the relationship between CX and student experience (SX) becomes evident. Students are a particular case of customers in the field of education. For this reason, it is possible to construct a definition of SX from the perspective of CX.

We have performed a systematic literature review of studies published during the past 10 years, from 2011 to 2020 (until June 2021), to identify SX definitions, factors that influence SX, and SX evaluation methods. In doing so, we focused specifically on undergraduate students. We have searched for articles that include "student experience" and "higher education" or "undergraduate" in their metadata (title, abstract and keywords) within five databases: Scopus, Science Direct, IEEE Xplore, Web of Science, and ACM Digital Library. A variety of SX definitions have been found. The results show that there are various scales

and surveys to evaluate SX. Furthermore, it is clear that SX is often evaluated in terms of satisfaction. As there is no generally agreed SX definition, we propose a holistic approach from a CX point of view.

The remainder of this paper is organized as follows. Section 2 introduces the theoretical background of this study. Section 3 presents the research method, research biases, research questions, search strings, exclusion and inclusion criteria, and a quantitative analysis of the results. Section 4 presents the selected studies, the results obtained, and answers to the research questions. In Section 5, we focus on answering the research questions, expressing our point of view. Finally, in Section 6, we present the conclusions of the review and propose an SX working definition.

## 2. Background

### 2.1. User Experience

User experience (UX) is the result of subjective perceptions of the use of goods and services in a given context. The ISO standard 9241-210 defines UX as follows: "person's perceptions and responses resulting from the use and/or anticipated use of a product, system or service". UX incorporates elements such as emotions, personal beliefs/preferences, perceptions, physiological/psychological responses, behaviors and the achievement of activities that occurred before, during, and after the use of a product or service [3]. UX is a concept closely related to CX, which extends the analysis of user interactions with a single product, system, or service to all interactions with a brand, through the products, systems and services that it offers. In this way, the concept of the user gives rise to the concept of the customer. The concept is relevant in our study given that, as a client, students are the users of services, products and systems offered by higher education institutions (HEIs).

### 2.2. Customer Experience

Although the term CX has been widely discussed, it does not have a clear, standardized definition. CX has been considered as "the physical and emotional experiences occurring through the interactions with the product and/or service offering of a brand from point of first direct, conscious contact, through the total journey to the post-consumption stage" [4].

Gentile et al. (2007) consider that CX has six dimensions: emotional, sensorial, cognitive, pragmatic, lifestyle and relational [5]. The emotional component involves the affective system, through the generation of moods, feelings, and emotions. The sensorial component involves stimulation, which affects the senses. The cognitive component is involved in thinking and conscious mental processes. The pragmatic component is involved in the practical act of doing something. The lifestyle component is related to the values and the beliefs of a person, and the adoption of certain lifestyles and behaviors. The relational component involves the person, social context, and relationships with other people.

Other authors such as Lemon and Verhoef, have defined CX as "a multidimensional construct focusing on a customer's cognitive, emotional, behavioral, sensory, and social responses to a firm's offerings during the customer's entire purchase journey" [6]. This definition makes it clear that the relationship between customers and companies exceeds the threshold of merely physical. Additionally, it mentions the concept of a customer's "journey", which is of great importance when analyzing CX.

Touchpoints are vital elements for analyzing customer's journeys. They have been defined as a representation of: "( . . . ) a specific interaction between a customer and an organization. It includes the device being used, the channel used for the interaction, and the specific task being completed." [2].

### 2.3. Student Experience

Considering that higher education students constantly interact with products, services and systems provided by educational institutions, their role as customers is evident. Thus, for the purposes of this work, we refer to SX as a particular case of CX, where students are customers of educational services, systems and products.

By understanding the dynamics of student interactions and their impact, it is possible to enhance the quality of their experiences and improve their satisfaction and wellbeing. To achieve this, it is necessary to identify the elements/dimensions/factors that compose the SX. In addition, it is important to know that HEIs are trying to evaluate SX.

One of the objectives of this systematic bibliographic review is to know precisely what definitions of SX have been adopted in the literature, as well as the factors and dimensions with which they have been developed.

## 3. Research Method

This literature review was performed based on the framework for literature review in software engineering proposed by Kitchenham [7], which includes the following three phases:

- Planning the review, which includes the process of defining research questions and strategies to carry out the review.
- Conducting the review, which includes the specification of inclusion/exclusion criteria, data extraction and article selection.
- Reporting the review, which requires the presentation and discussion of the results.

To ensure the quality of the review, we have incorporated the checklist of elements proposed by the PRISMA methodology, focusing on improving the realization of systematic literature reviews and meta-analysis [8].

### 3.1. Research Questions

We oriented the systematic literature review to the following topics: (1) SX definitions, (2) dimensions/attributes/factors regarding SX, and (3) methods that are used to evaluate SX. Table 1 presents the three research questions (RQ) that guided our study.

**Table 1.** Research questions used in the review.

| ID | Research Questions (RQ) |
| --- | --- |
| RQ1 | What is (undergraduate) SX? |
| RQ2 | What dimensions/attributes/factors influence (undergraduate) SX? |
| RQ3 | What methods are used to evaluate (undergraduate) SX? |

### 3.2. Literature Search

In this review we examined the literature published within the last 10 years (from 2011 to June 2021), indexed in five databases: Scopus, Web of Science, Science Direct, IEEE Xplore, and ACM Digital Library. In all five databases we searched for the following keywords: "student experience" and ("higher education") or "undergraduate". The number and percentage of studies available in each database are listed in Table 2.

**Table 2.** Search results in databases using the keywords.

| Database | Number of Studies | % Studies |
| --- | --- | --- |
| Scopus | 3114 | 71.1% |
| Web of Science | 800 | 18.3% |
| Science Direct | 390 | 8.9% |
| IEEE Xplore | 59 | 1.3% |
| ACM Digital Library | 15 | 0.3% |
| Total | 4378 | 100% |

### 3.3. Study Selection Criteria

We searched for studies focusing on SX, in all areas, including all types of research methods. The inclusion criteria are presented in Table 3. As indicated in the table, the study was limited to SX in higher education, at the undergraduate level, excluding highly

technical applied cases (such as the analysis of the student's experience in highly specialized work environments), and strictly pedagogical articles. This is because SX integrates multiple elements of the student's life (such as culture, emotions, education, commitment), and, therefore, should be approached from a holistic perspective.

**Table 3.** Study selection criteria for this review.

| ID | Category | Criteria |
|---|---|---|
| IN1 | Inclusion | Articles published between 2011 and 2021 |
| IN2 | Inclusion | Articles referring to research questions |
| EX1 | Exclusion | Studies focused on levels other than undergraduate |
| EX2 | Exclusion | Strictly pedagogical articles |
| EX3 | Exclusion | Highly technical applied cases |

Given the relevance of some articles referenced by those selected, we decided to also include them, despite not meeting the search criteria expressed in this review.

### 3.4. Study Selection

By applying the selection criteria, we gathered 65 studies. Figure 1 shows the search and selection process flow, applying the inclusion and exclusion criteria in each step, and removing duplicates.

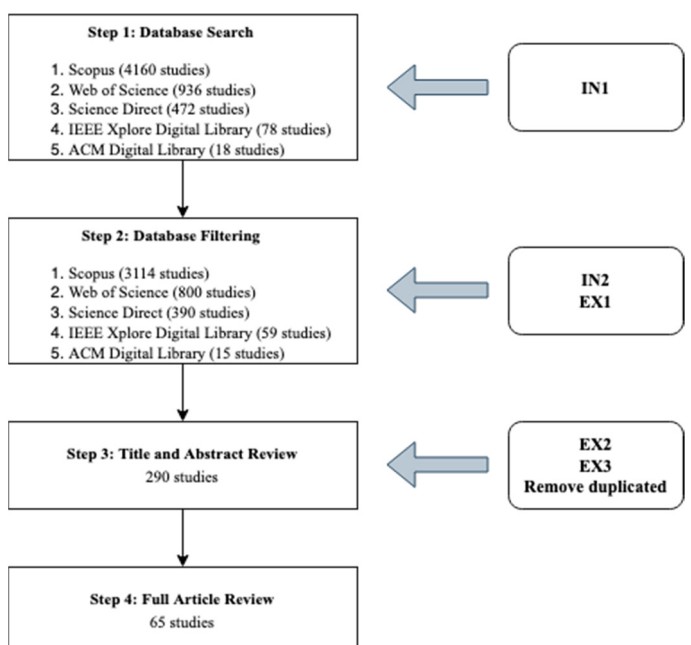

**Figure 1.** Flow chart with the results of the study selection process.

## 4. Data Synthesis

The review includes the data synthesis of the 65 studies selected for a full review. This analysis consists of the summarization and tabulation of the studies, identifying: (i) publications over the years, (ii) document type, and (iii) subject area.

### 4.1. Year of Publication

The distribution of documents published over the years shows continuous growth. This highlights the increasing interest in the research on SX. Additionally, Figure 2 shows that, although the year 2021 is in progress at the time of this review (June 2021), it is possible to forecast that the increasing trend will continue.

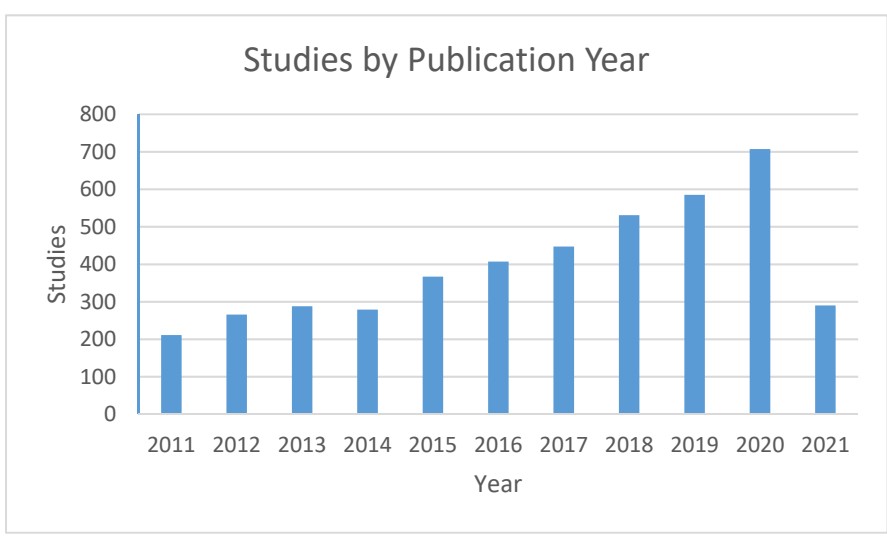

**Figure 2.** Studies published by year.

*4.2. Document Type*

We analyzed the document types of the studies reviewed. Documents are classified into books, book chapters, reviews, conference papers, and journal articles. The vast majority of the 65 studies selected were journal articles (53), followed by book chapters (6), conference papers (3), and reviews (2). Books had the smallest number of documents (1). Figure 3 shows the document type distribution of selected studies.

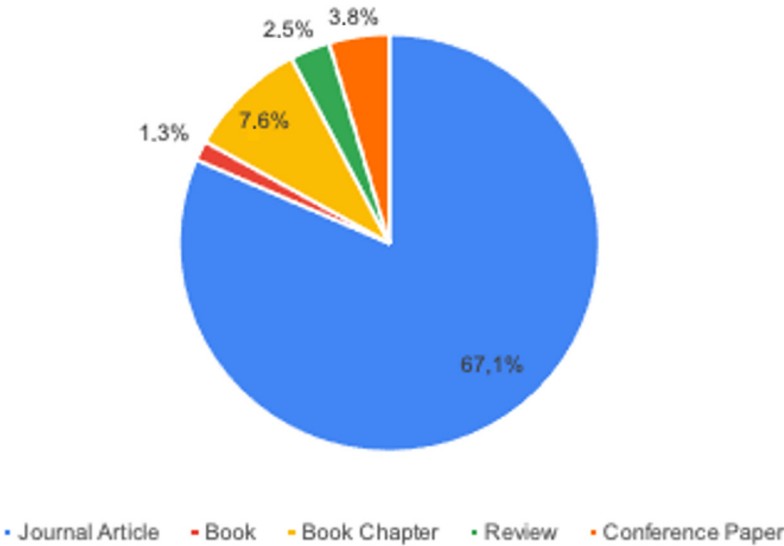

**Figure 3.** Document type.

*4.3. Subject Area*

Figure 4 indicates the distribution of the selected studies according to their subject area; the six subject areas are the ones that databases are using when categorizing the documents. The vast majority of studies are in the area of "Social Sciences" (40), while the areas with the smallest number of studies are "Tertiary Education and Management" (1), and "Education/Educational Research" (1). We point out the importance of the considerable number of studies with a multidisciplinary approach (19), which highlights the need for a multisectoral approach to SX.

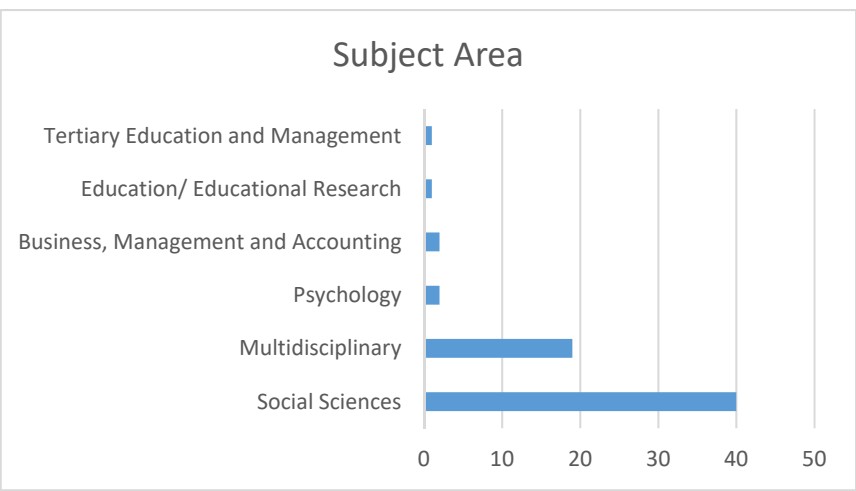

**Figure 4.** Studies subject area.

## 5. Answering the Research Questions

### 5.1. RQ1: What Is (Undegraduate) SX?

The SX concept is relatively new [9]. The concept emerged driven by a fundamental change in the higher education sector, where teaching centered on the student and oriented toward learning processes began to be prioritized. However, over the years, SX has been defined in multiple ways [10]. This is not a minor conceptual problem.

Tan et al. pointed out inconsistencies when defining the SX concept [11]. The "experience" definition is, on the other hand, diffuse and, in many cases, the term is, "as the concept of value", rarely well defined [12–14]. Despite the problems of inconsistency, the SX concept makes a clear reference to the experiences of students. Even so, it is sometimes difficult to distinguish between such experiences and students' everyday life experiences [15]. SX has become a topic of great interest, especially in the past decade (2010's), in academic and learning environments [16]; very few studies have focused on the students' holistic experiences [17].

This concept has been widely used in universities for advertising purposes. In this way, universities offer an "excellent student experience" in order to attract consumer/students [18]. SX shows a trend in the use of the concept as an indicator of quality and/or satisfaction. Authors such as Farhan and Carey have pointed out that, since the focus of HEIs is on student satisfaction, it is possible to observe consumer attributes in students [19,20].

Since the emergence of the concept, it has been referred to as a strictly pedagogical field. This has changed over the years. Baranova et al. have observed how the conception of the student experience has evolved, including, in addition to the learning and teaching experience, the interaction of students with the administrative and support services of HEIs, outside the classroom [21]. Douglas et al. defined SX within the academic sphere as the "experience of higher education teaching, learning and assessment and their experience of other university ancillary service aspects" [22]. Other examples of the definition of the concept outside the classroom are those of Harvey and Knight, and Arambewela and Maringe, who use the term "total student experience" to refer to experiences lived by students in contexts broader than the physical place of learning [23,24].

It is important to point out that the learning experience, being related to the learning outcomes of the students, directly influences students' academic success. Kahu et al. also observed how the academic success of students, as well as the awareness of that feeling, influence student engagement, as well as other factors that we have covered in this section [25].

Pötschulat et al. analyzed the SX concept from a sociological point of view. They concluded that the use of the concept tends to be a political intervention, in view of its sociological inadequacy. They also point out that the use of the concept smuggles in a

market logic that homogenizes the elements in which it could be composed [26]. We believe that not only a poorly focused market logic can homogenize the elements of the student's experience, but also any superficial approach to the concept. The blurring of the elements that compose SX can make it difficult to evaluate.

Detaching the concept from the academic field, Baird and Gordon have said that the SX concept refers to the various experiences of subjects while they are students. Similarly, Temple et al. defined SX as the totality of a student's interaction with the institution, contemplating the interactions carried out outside the merely academic field [27,28]. Rahim and Jusoh mentioned that the term SX refers to the "student learning and living development at higher educational institution starting from early admission to the university until graduation" [29].

The trend towards a broader approach to the SX concept is evident when we observe new concepts that involve experiences in different contexts. An example of this are the terms "student transitions" [30,31], "trajectories" [32] or "pathways" [33], which make clear reference to the students' journey.

Despite the focus on teaching and learning in which SX has been addressed, there is evidence in the literature that focuses on students as customers [34–37]. Within the less typical views in the market, students have been analyzed as active customers [38], citizens [39], and coproducers [40,41]. In this way, the active role of students comprises student–academic collaboration relationships for the production of knowledge, and student–society relationships, where the student is analyzed as part of society and the relationship of convenience between them is ethically analyzed. Part of the research carried out in this matter has addressed a vision of the education market dealing with the discourse of the value of cocreation, where students contribute their capacities to the construction of the educational system [42–45]. This point is important because, as a result of the interaction of students with HEIs, it is possible to create value. This highlights the mutual benefits of the student–HEI relationship. The value of cocreation by the customer has been analyzed by Prahalad and Ramaswamy, and identified as the experience of a specific customer, at a specific point in time and location, in a specific interaction context [46]. As a result of students' value creation, their role as a customer is once again evident. Gupta and Vajic have related customers' experiences with the knowledge acquisition process, saying that "an experience occurs when a customer has any sensation or knowledge acquisition resulting from some level of interaction with different elements of a context created by the service provider" [47].

It should be noted that there have been criticisms of the public policies that have introduced a market logic into education and have placed the student as a mere customer [48,49]. Research that criticizes market logic focuses on two aspects. First, the impact of new management practices on academic work and identity [50–52]. Second, the negative impact on students' expectations and experiences, since there has been a transition from student to learner, where they are pushed to take responsibility for their own learning as independent, self directed individuals [53–56]. Jæger and Gram analyze the Chinese and Danish exchange student context, pointing out marketization in education approaches; national and transnational discourses, and government and institutional policies are factors that influence consumer orientation among students [57]. The economic model by which HEIs are financed does not only include the factors that affect consumer orientation.

For us, and like Dollinger and Lodge, the large quantity and variety of definitions of the SX concept denotes an evident diversity of the conceptualization of experience in higher education [58]. This undoubtedly represents a problem, considering that in the literature a direct relationship between SX and student satisfaction is made, and satisfaction is pointed out as an indicator of quality in HEIs [23,59], understanding satisfaction as the positive difference between expectation and perceptions. In this way, it is possible to see how various evaluation mechanisms (such as SERVQUAL and National Student Survey (NSS)) have been used to assess SX, which, according to multiple authors, is not sufficient or adequate for this task, in determined contexts.

Table 4 indicates the different approaches to the SX concept and the associated studies.

**Table 4.** Studies approaches to the SX concept.

| Approach | Studies |
| --- | --- |
| Academic | [16,21–25,29] |
| Student Journey | [30–33] |
| Sociological | [26] |
| Holistic | [15,17,23,24,27,28,58] |
| Customer | [18–20,34–41,43–45,48,57] |
| Market Logic | [48,50–57] |
| HE Quality | [23,59] |

*5.2. RQ2: What Dimensions/Attributes/Factors Influence (Undegraduate) SX?*

The concept of SX is treated in a wide number of studies using different approaches, depending on the specific goals of each research. As Mieschbuehler mentioned "the term student experience can be found everywhere" [18]. In addition, a predominant approach to learning and teaching was observed, focusing on the student learning experience and learning outcomes.

After conducting an exhaustive analysis of the studies in the literature, we can indicate that the studies usually focus on three topics: (i) on learning and teaching, where the methodological and technological aspects that allow increasing learning outcomes and facilitate teaching are analyzed, including evaluative processes; (ii) on engagement/commitment, where the sense of belonging, partnership and engagement/commitment between the students and the HEI are analyzed; and (iii) on the wellbeing of the students, where the sensations and feelings of the students are analyzed, as well as their overall wellbeing and interpersonal relationships. This is similar to that observed by Bista, who mentions that the perspectives of international students' experiences include sociocultural identities, contextual influences on their learning experiences, their wellbeing experiences, and their poststudy experiences [60]. Bates et al. have identified main themes referring to students' experiences, such as "Learning Environment", "Work–Life Balance" and "Wider University Community" [10].

Kahu and Picton have analyzed first year students' experiences using photo elicitation. They analyzed the information provided by the students thematically and categorized it, focusing on three dimensions of their experience: life, university and learning [61]. Other authors have highlighted the use of surveys focused on students' experiences, such as the Australian Student Experience Survey (SES) [62,63]. This survey categorizes its questions into five conceptual groups related to satisfaction: learner engagement, teaching quality, learning resources, student support, and skills development [64]. Additionally, a study by Hong et al. analyzed the trends in which the student experience was addressed. These trends are "First-year transitions", "Student engagement and outcome", "Student experience in learning", and "Quality management and service satisfaction", among others [65]. Tan et al. realized, through a literature review, the prevailing research streams when speaking of "student experience". These are "exploration of learning experience", "exploration of student experience", "gender differences in assessment of higher education experience", "improvement in quality of student experience", and "student satisfaction with higher education experience" [11].

Hailikari et al. observed that the focus on learning and teaching in SX is the most common, generally referring to strictly academic environments and factors. For good reason, they mention that the experience of the students is affected by more factors [66]. Furthermore, student learning experience in HEIs is also a multifactorial issue that is affected by environmental variables, such as the HEI's infrastructure, classroom quality, residential facilities, IT facilities, and sports facilities [67]. Ruhanen et al. analyzed students' experiences in a tourism immersion internship. In their work, it was observed how the

expectations, experiences and satisfaction of the students are directly related to their learning experience [68].

From the point of view of technological support, Barret et al. addressed the role that learning technologies and learning management systems play in improving students' experiences. This is evidenced in the learning outcomes of students [69]. Likewise, Yu and Bryant analyzed the use of technology in the learning experience of students, and the consequent impact on their lives. Although the impact of technology on SX and on learning outcomes is notable, the lack of studies linking these elements is evident. As Yu and Bryant said "we know little about how students' personal, digital and educational lives intersect and shape their learning experience" [70].

The prevailing focus of teaching and learning on student experience has led to the development of tools that can increase student satisfaction. This is the case of the User eXperience Design for Learning Honeycomb framework, which aims to create valuable online learning experiences. Troop et al., analyzed the use of this framework and detected four aspects in the online platform's design that students find valuable: accessibility, usefulness, intuitiveness, and desirable [71].

There are surveys that allow us to get notions about the learning experience of students, more specifically, their learning outcomes. Douglass et al. used, in their study, the Student Experience in the Research University Survey (SERU-S), which allows accountability for assessing and reporting learning outcomes in higher education [72].

Students' engagement in relation to students' experiences is one of the most frequently observed themes in the literature regarding SX. Student engagement has been associated with other elements of SX, such as student learning and achievement [73]. Studies carried out with the Student Experiential and Engagement Value Index (SEEVI) have made clear the close relationship between students' engagement with HEIs, and their experience [74]. Student engagement is influenced by many factors, such as sense of belonging, institutional commitment, student satisfaction, and the quality of the students' experiences [75]. Chambers and Chiang have used the National Survey of Student Engagement (NSSE) to analyze students' perceptions about the issues that challenge them to develop skills, awareness, confidence, and other factors that influence their engagement. The close relationship that exists between engagement and the experience of students is evident [76].

Meehan and Howells analyzed the impact of students' experiences in HEIs and their sense of belonging. In their work, they concluded that "the students' states of being, belonging and becoming are critical to understand in order to engage students for a successful experience in higher education" [77]. It is evident that the sense of belonging and the engagement of the students are closely linked, and that sense of belonging plays a vital role in the wellbeing of the students. Similarly, Naylor et al. highlighted the importance of students' sense of belonging while reducing their stress levels, to improve their experience. This also reduces the risk of dropping out [78]. Other authors directly relate student retention with increasing students' satisfaction by improving their experience [79].

Both sense of belonging and workload stress have been validated, together with feeling supported and intellectual engagement, as elements that serve to evaluate the experience of students. In this way, the crucial role that the affective aspects of students have, in terms of the quality of their experiences, is revealed. In addition, student engagement has been related to student happiness, which has implications for students' experiences and overall wellbeing [80].

Grebennikov and Shah, monitoring trends in student satisfaction, have realized the importance of satisfaction in student engagement. If students had good and satisfactory experiences, they were more reluctant to drop out of their studies. In this way, the relationship between the wellbeing of students and engagement is evident, which is one of the focuses most approached in the literature regarding SX [81].

Dominguez-Whitehead has analyzed the role of nonacademic support services in SX, considering that students' experiences extend beyond the academic context. The analysis of nonacademic services is related to personal, institutional, and interpersonal factors.

Investigating the relationship between students' experiences with nonacademic services can offer alternatives for rectifying problems and concerns related to nonacademic service provision, improving students' experiences and wellbeing [82].

Smith and Segbers analyzed the impact of interculturality on students' experiences [83]. This is important, because through good practices that address intercultural education situations, financial, educational, and cultural benefits can be achieved. It is also important to mention authors who have pointed out cultural differences in learning, which certainly affects the perception of quality in HEIs and the experience of the students [84].

Tan et al. observed that gender differences in the assessment of higher education experience is a common research stream in studies that analyze students' experiences [11]. In this way, it is evident that the experience of students is subject to factors specific to individuals, such as their own gender. Similarly, Jayadeva et al. observed that students' perceptions of higher education vary depending on factors specific to individuals and their environment, such as national contexts, and may be mediated by factors such as educational quality perceptions [85].

Another example of student experience assessment under specific contexts is observed in the work of Brady et al., who developed a survey to analyze the perspectives and experiences of STEM (science, technology, engineering, and math) students in historically black colleges and universities (HBCUs) [86]. African American students' experiences in HBCUs can differ significantly from those in other communities, and in other HEIs, considering the racial segregation that has taken place in the USA over the past century. It has been observed that some HEIs outperform others because of the alignment between the values and aspirations of the students with those exhibited by their HEI [62].

It is important to mention that the experiences of students who transition to higher education turn out to be an important factor in terms of their learning outcomes and engagement [87]. As the experiences of first year students have such a significant impact on their perception and academic results, it is possible to explain the large number of studies focused on this group of students [26,61,77,78,88,89].

SX has been addressed in a large number of studies using different approaches. While these approaches are well suited to providing a satisfying experience, and can complement each other, factors unique to the student's life should not be omitted. Cheeseman has considered that the experience of students does not strictly refer to the relationship between students and their HEI, but encompasses aspects of the students' day to day lives [90]. Students' personal experience, familial experience, institutional experience, national milieus, class, ethnicity, gender, race, religion, sexual orientation, or responsibility for dependents, are factors that influence SX [54,91–93].

We share Sabri's critique on the conceptualization of the "student experience" as a means of discriminating between the value of different educational experiences [48]. We believe that this occurs due to the multiple factors inherent to the student's life that are overlooked, either by the surveys or the research approaches.

In view of the studies analyzed in this review, we categorized the factors that influence the student experience into three dimensions. To achieve this, we analyzed the approach of the studies referring to SX, the conceptual groups attended, and the elements considered in evaluation scales that address the subject. The results shown in Table 5 describe the observed dimensions of SX and the factors related to each dimension. In addition, studies related to these factors are included.

**Table 5.** Student experience observable dimensions.

| ID | Category | Criteria |
|---|---|---|
| Social Dimension | Community Relationship | [21,58,67,70,82,88,91,94–97] |
| | Institutional Engagement | [25,60,67,73–79,81,86,88,89,94–96,98–105] |
| Educational Dimension | Learning Engagement | [17,21,25,29,61,66,67,73–76,78,79,81,86,89,95,104–107] |
| | Higher Education Quality | [11,24,34,37,58,59,63,76,81,84,98,108–113] |
| | Learning Resources/ Learning Environment | [10,16,35,62,65,66,69–71,73,75,91,99,105,107] |
| | Educational/ Support Services | [60,69,78,82,97,102,106,114] |
| Personal Dimension | Student Development and Outcomes | [17,29,67,72,76,95,114] |
| | Student Feelings and Emotions | [10,67,70,78,80,95,97,115] |
| | Environment Relationship | [26,58,76,90,91,95,96,105] |
| | Student Thoughts, Identity and Background | [10–12,15,17–20,22–24,26–28,30–41,43–45,48,50–58,60–63,65–69,71,76,77,79,81,83–93,95,97,99–102,105–108,114,115] |

In the social dimension, we found factors related to the interaction of students with the various actors and entities that provide services in HEIs. This includes relationships with educational staff, HEIs helpdesks, classmates, and tutors, among others. In addition, this dimension includes students' institutional engagement, as this is generated mainly from interactions between entities.

In the educational dimension, we found factors related to the learning experience of students, which determine their learning outcomes and success. In this way, aspects such as teaching quality, learning resources, student educational support and learning engagement are included.

In the personal dimension, we found factors related to aspects of the life of each student. Some factors in this dimension are difficult to evaluate using standardized tools. This dimension includes aspects such as gender, socioeconomic level, cultural background, state of health, emotions, skill development, and day to day experiences, among others. The emotions experienced by students are particularly important, as they affect short and long term adaptational outcomes in higher education [116].

It should be noted that, sometimes, the factors mentioned above do not refer to having an effect on a single dimension of the student's experience but may also affect factors typically associated with other dimensions.

*5.3. RQ3: What Methods Are Used to Evaluate (Undergraduate) SX?*

Evaluating SX is of growing interest among various governments and HEIs. As a result of efforts to improve student satisfaction, frameworks and scales have been proposed to measure students' satisfaction, and overall student experiences in HE. Otherwise, student experience is typically measured using policy driven metrics [10]. This is the case with tools such as the NSS [117], NSSE [118], SES, Course Experience Questionnaire (CEQ), and variations of the SERVQUAL scale, to evaluate satisfaction with educational services in different case studies. In addition, some tools have focused on engagement aspects, such as SEEVI [63].

All the tools mentioned above have in common the characteristic of gathering opinions directly from students. While this is a positive aspect, it is hardly sufficient. Nair et al. have pointed out the importance of the commitment of the university, faculty management and academic staff to address students' issues correctly [108]. In this way, it is evident that the various mechanisms to evaluate the experience of students must be matched with appropriate measures to effectively improve student satisfaction.

The term "student experience" is usually associated with students' perceived satisfaction. In the words of Meehan and Howells, "measuring student experience in terms of satisfaction is a national measure used by prospective students when considering their higher education choices" [88]. Ammigan and Jones have analyzed the degree of satisfaction that international students perceive in the different dimensions of their university experience. By performing a regression analysis on the data obtained from the students,

they were able to identify four dimensions of satisfaction related to students' experiences: arrival experience, learning experience, living experience, and support services experience [106]. To carry out their study, they relied on the International Student Barometer (ISB), which tracks and compares the decision making, expectations, perceptions and intentions of international students, from application to graduation [119].

Student satisfaction and student experience are often measured in terms of the quality of services or products that HEIs offer to them [94]. Additionally, the quality of the services offered is usually measured in terms of teaching [98,109,110]. In view of the dimensions that we have analyzed in this article, it is evident that a quality approach focused solely on teaching is a mistake.

The relationship between student experience, quality, and satisfaction is more evident when we analyze the relationship between students' experiences and university accreditation, carried out by agencies such as the Quality Assurance Agency (QAA) in the UK [111]. Camgoz-Akdag and Zaim have worked with the five dimensions of SERVQUAL as a basis for proposing a conceptual model of student satisfaction. In this study, the quality variables perceived by the students were related to their satisfaction and, thus, to their overall SX [112].

SERVQUAL [120] is one of the most widely used scales to evaluate students' perceptions of the services delivered at HEIs. Even so, this scale does not seem to be efficient in certain contexts, which is why many studies modify this service quality scale depending on their objectives. Vaughan and Woodruffe-Burton tested a disabled user specific service quality model (ARCHSECRET) [113], as opposed to a modified version of SERVQUAL, in the context of disabled students. In practice, this model turned out to be more efficient in the specific case of students with disabilities [114]. For cases such as the previously mentioned, it is valid to assume that it is not possible to develop a single model that is capable of evaluating the perceptions and satisfaction of students, without disregarding elements of their personal lives.

We found several studies aimed at measuring student satisfaction, taking into account rather "unusual" elements. Shahijan et al. have examined the experiences of international students' satisfaction with HEIs. In their work, they focused on measuring the experiences of international students, as well as their satisfaction level and perceived performance [99]. This is a complex task, considering that the international student experience is a multi-dimensional issue influenced by various personal and cultural factors [91]. Meehan and Howells conducted a study to analyze the perceptions of first year students, regarding students' experiences and satisfaction. For this work, they used the Student Experience Evaluation (SEE) instrument and the National Student Survey (NSS). Their analysis identified three elements that influence the students' experiences and satisfaction: the academic staff they work with, the nature of their academic study, and their sense of belonging [88].

Satisfaction measurement extends to the use of more scales than just SERVQUAL. This is the case in studies that have used the UK NSS, which is very popular [63]. This is usually considered in the student evaluations of teaching (SET) cases of study. However, this survey was also criticized. Zaitseva et al. have used concept mapping software in their studies to aid in the interpretation of the qualitative data from student satisfaction surveys. In this way, they were able to identify aspects of SX beyond the survey questions. In their studies, they have mentioned the importance of being cautious with statements that imply that survey results are representative of the entire student experience [95]. For her part, Sabri criticizes the role of the NSS, arguing that it has nuances that cause the tool to exceed its validity or intended use [96].

Although student experience is usually measured in terms of student satisfaction, multiple governments and organizations do so in terms of the quality of products and services provided by HEIs using quality indicators (e.g., sufficiency of learning resources, classroom design, and achievement of learning outcomes) [37].

For their part, Naylor et al. have defined, in their work, four scales to measure students' experiences at university: "belonging", "feeling supported", "intellectual engagement", and

"workload stress" [78]. These scales can be related to some of the dimensions in which we have observed that the student experience is analyzed, mentioned in Section 5.2, such as "Learning Engagement", "Student Feelings and Emotions", and "Educational/Support Services".

It is interesting to mention the work of Tan et al., in which it was observed that one of the major streams of research on student experience quality in HE refers to student satisfaction with the higher education experience [11].

Thuy and Tao analyzed the impact of students' experiences on brand image perception. Unlike most studies, they adopt the experiential marketing view, which uses the SERVQUAL scale as the basis for measuring students' evaluations of HEI products or services. In their work, students' experiences attributes related to brand image perception, such as in classroom sensations and behaviors, affective experiences, and relationships, were analyzed [100].

As mentioned in Section 5.2, the factors that organizations focus on when analyzing the experience of the students are institutional and learning engagement. These are commonly evaluated together, with the related "student experience" to be used as a reference for future students regarding "good" places to study [101]. Thus, the focus on engagement is usually presented as a market focus. Regarding engagement, several studies have used their own measurement scales. This is the case in the work of Musa and Saidon, where the Student Experiential and Engagement Value Index (SEEVI) is used [74]. In this way, it is evident that the use of different measurement tools, with different approaches to students' experience, can be particularized to the analysis of specific SX factors. Similarly, Chambers and Chiang have studied how students perceive elements that challenge them to develop skills, awareness, and confidence. To accomplish this, they utilized the NSSE. As a result of their work, information has been provided to improve students' experiences, focused on three categories of high importance: "Academic experience"; "Social experience", and "Campus environment" [76].

Khalifa et al. studied the perception and engagement of higher education students of and with the various tangible and intangible services provided by their HEIs. For this work, they used the College Student Experiences Questionnaire (CSEQ) [121], which is intended to measure the quality of SX. The findings of this study show that students with disabilities experience less satisfaction with intangible services. This shows how the experience and perceptions of students vary depending on factors inherent to the students' lives [102].

Beck and Milligan investigated the factors that influence the institutional commitment (IC) of students, especially in online education. To do this, they relied on the College Persistence Questionnaire (CPQ) [103]. In their work, aspects of students' experiences were identified as predictors of IC [104]. In this way, the existence of a relationship between the SX and the IC is clear. In the work of Barrie et al., the development of an SX survey was reported, which highlights the aspects that influence SX. In the authors' words, this survey helps HEIs staff to "identify the extent to which student engagement is maximized" [105]. This again shows the relationship between students' experiences and their engagement.

Moss and Pittaway analyzed the importance of student expectations and engagement regarding HEIs during their transition stage. For this, they analyzed the student journey by identifying key touchpoints for online students in transition [89].

The holistic approach to SX has the drawback of being more difficult to assess. As we incorporate more elements to ensure an adequate approach to genuine students' experience, the analysis becomes complex. If we add to this problem the specificity of each student journey, we find ourselves, according to Heron, in a situation in which no survey or questionnaire can give a certain account of the genuine experiences of students [97].

Dollinger and Lodge, analyzing the value in SX, mention that the student satisfaction measurement should be holistic, and not one dimensional, or analyzed only at the "end of an experience" [58]. In their study it was determined that the student–staff partnerships may enhance value in the students' experiences. This statement complements the ideas that we have raised in previous sections, where the interpersonal relationships of students (both

inside and outside the classroom) directly contribute to their engagement and satisfaction. In this way, the overall student experience is improved.

Bevitt mentioned the need for an innovative assessment of student experience in higher education. For this, and in accordance with our judgment, it is necessary to use a holistic approach when evaluating SX. According to the author, the "assessment for student experience must be viewed as a complementary layer within a complex multi-perspective model of assessment" [107].

In our opinion, the process of evaluating SX has two major difficulties. First, students' perceptions are unique, personal, and subjective, which is why SX is difficult to analyze using standardized tools. Second, a student's journey is very broad and complex, and many dimensions must be evaluated. We believe that two levels of evaluation should be used. The first level should focus on the very nature of the touchpoints, to more accurately assess the perceptions of the students, considering a mixed approach (quantitative and qualitative). It should examine a student's interaction with specific services, products and systems, such as educational services, the use of information technologies for learning, and psychoeducational support services. The second level of evaluation should focus on the holistic experience of a student, considering personal aspects of the individual; scales such as SERVQUAL usually ignore cultural and/or emotional aspects. An article indicates the complexity of CX evaluation [122].

As we have observed, SX is often evaluated in terms of satisfaction. A complete SX evaluation would allow HEIs to improve the quality of their services. The instruments used for this work, which generally use self reported methods, can encourage universities to adapt and improve their service, shaping their courses and changing the way they relate to students [81].

Table 6 shows the different evaluation instruments and the studies in which they were mentioned.

**Table 6.** Instruments to evaluate SX aspects.

| Evaluation Instrument | Studies |
| --- | --- |
| National Student Survey (NSS) | [63,88,96] |
| National Survey of Student Engagement (NSSE) | [76] |
| Student Experience Survey (SES) | [62,63] |
| Student Experience Evaluation (SEE) | [88] |
| Student Experiential and Engagement Value Index (SEEVI) | [74] |
| International Student Barometer (ISB) | [106] |
| SERVQUAL | [100,112,114] |
| Course Experience Questionnaire (CEQ) | [63] |
| ARCHSECRET | [113,114] |
| College Student Experiences Questionnaire (CSEQ) | [102] |
| College Persistence Questionnaire (CPQ) | [103,104] |

## 6. Conclusions and Future Work

The term "student experience" is widely used in the literature, but unfortunately there is currently no general consensus on the term. For this reason, it is possible to find a large variety of definitions with specific approaches related to the specific purposes of each study. In addition, it has been criticized that some works using the SX concept are theoretically poorly developed [12,14]. It is worth mentioning that, in this study, we have focused on the experience of undergraduate students.

In general, the term SX is used with academic approaches and refers to the experiences of students in the context of learning and teaching, that is, learning experiences. However, it is possible to find a diversity of approaches in the literature. Among the articles that we have analyzed, the experience of students is conceived as the totality of their perceptions and interactions with different educational services, staff, and peers, while they are students.

In this study, we included some documents that did not meet some of the selection criteria. This is because some of these articles are important for conceptualizing SX as a relatively new concept [9], a topic of great interest [16], and define SX both within the academic sphere [22], and with a holistic perspective [23,24]. Other articles are relevant to understanding the student as a consumer, since they contextualize the student as a coproducer [40–42]. Other documents are useful for understanding some aspects of the student's life that influence SX, such as culture, day to day aspects, and emotions [92,93,116].

We found articles that analyze the SX concept from a customer experience approach, where the student is considered a consumer of educational products and services. This approach has been criticized, since it has been pointed out that the introduction of a market logic in higher education can homogenize the elements that compose the student's experience [25].

In view of the articles analyzed in this review, we categorized the factors that influence the students' experiences in three dimensions, analyzing the focus of the articles, the concepts covered, and the elements considered in the associated evaluation scales. The analyzed studies focused mainly on three aspects, learning and teaching, student engagement/commitment, and student wellbeing.

The observed dimensions were: (i) social dimension, which comprises relational aspects between students and the various actors with whom they interact throughout their university life, i.e., educators, HEI staff, classmates, service assistant; (ii) educational dimension, which includes factors that influence the student learning experience, i.e., learning resources, student educational support, teaching methodology; and (iii) personal dimension, which includes aspects of the student's life, i.e., sex, socioeconomic level, cultural background, health status, emotions, feelings, skill development, and day to day experiences. Within the personal dimension, it can be observed that very few articles that deal with the SX concept directly address students' emotions. This is a problem, considering that customer emotions are a critical part of the CX. Added to this is the problem that emotions and feelings are not considered as variables to estimate customer satisfaction [115,123].

SX evaluation is usually carried out in terms of students' satisfaction and uses self-reported methods to collect the required information. Scales such as NSS or a modified version of SERVQUAL for specific contexts are used. In addition, it is important to mention that a variety of instruments are used to evaluate aspects of the student's experience, such as the NSSE or the SEEVI. Surveys and questionnaires used to evaluate SX are frequently related to the quality of education and HEIs. This highlights the role of aspects such as student satisfaction in quality assurance organizations. Obviously, if students have good experiences, their satisfaction will increase.

The SX evaluation process is very challenging. This is due to the complexity of the student journey, and the fact that students' experiences are personal and are made up of subjective perceptions. For this reason, we recommend that SX evaluation should be performed at two levels; the first level should focus on the nature of the touchpoints, and the second one should focus on the holistic analysis of the student's journey.

In our view, the SX concept refers to all the physical and emotional perceptions that a student or future student experiences in response to interaction with products, systems or services provided by a HEI, and interactions with people related to the academic field, both inside and outside of academic spaces. The interactions mentioned in this definition contemplate aspects that influence the development of students such as culture, sexuality, physical disabilities, among others.

**Author Contributions:** Conceptualization, N.M., C.R. and S.C.; methodology, N.M., C.R. and S.C.; validation, N.M. and C.R.; formal analysis, N.M. and C.R.; investigation, N.M., C.R. and S.C.; data curation, N.M.; writing—original draft preparation, N.M.; writing—review and editing, N.M., C.R. and S.C.; visualization, N.M.; supervision, C.R.; project administration, N.M. and C.R. All authors have read and agreed to the published version of the manuscript.

**Funding:** Nicolás Matus is a beneficiary of the PUCV Ph.D. Scholarship 2021, in Chile.

**Institutional Review Board Statement:** Not applicable.

**Informed Consent Statement:** Not applicable.

**Data Availability Statement:** Not applicable.

**Acknowledgments:** The authors would like to thank the School of Informatics Engineering of the Pontificia Universidad Católica de Valparaíso (PUCV).

**Conflicts of Interest:** The authors declare no conflict of interest.

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
