# Peer review of "Student eXperience: A Systematic Literature Review"

_applsci, doi:10.3390/app11209543_

Round 1

Reviewer 1 Report

This research is a literature review of the student eXperience through the lens of the Customer eXperience.

The analyses are consistently carried out, and the methodology is strong.  (I wonder if the Document types other than Journal articles can be excluded as well, being just a minor part of the sample). The method of identification of the subject areas is not disclosed (lines 170-180), but generally related to the databases.

Limitations of the research are not identified. 

Overall, the research is very well designed and interesting. 

Author Response

Response to Reviewer 1 Comments

Point 1: This research is a literature review of the Student eXperience through the lens of the Customer eXperience. The analyses are consistently carried out, and the methodology is strong.

Response 1: We really appreciate your kind comments.

Point 2: I wonder if the Document types other than Journal articles can be excluded as well, being just a minor part of the sample.

Response 2: Indeed, references others than journal articles are a minor part of the sample. However, we think they provide useful insights.

Point 3: The method of identification of the subject areas is not disclosed (lines 170-180), but generally related to the databases.

Response 3: We appreciate your suggestion. Section 4.3 now indicates that the six subject areas are the ones that databases are using when categorizing documents. In fact, each database is using specific terms when classifying areas of knowledge; we tried to unify the 5 classifications in a rather subjective, but hopefully soundful way.

Point 4: Limitations of the research are not identified.

Response 4: We think that the main limitation of our study is its focus on (only) undergraduate level students. The updated version of the manuscripts indicates its focus in several parts of the document: abstract, introduction, research method, and conclusions.

Point 5: Overall, the research is very well designed and interesting.

Response 5: Again, we very much appreciate your kind comments.

Reviewer 2 Report

It may be more accurate to revise your research questions to align with the criteria utilized to identify studies for inclusion in the review (i.e., specifically, the focus on undergraduate SX, as your results may not be generalizable to graduate-level SX). 

It would also be a helpful inclusion to briefly explain why you chose to exclude pedagogical articles (which seemingly have the potential to align with RQ 1&2), particularly given the assertion that "Since the concept's emergence, it has been referred to as a strictly pedagogical field. " (line 200). It would also be helpful to operationally define "highly technical applied cases" and explain why they were excluded. There is no problem with your decision to exclude what you did - it would just be helpful to briefly explain your rationale for such choices. 

You note that "Given the relevance of some articles referenced by those selected, it has been decided to include them despite not meeting the search criteria expressed in this review" (lines 144, 145). It would be helpful to provide examples or describe the criteria that made such articles relevant enough to be included despite not meeting the described criteria. 

Your statement "We point out, like other authors, that the national and transnational discourses, government and institutional policies are factors that influence consumer orientation among students" (lines, 258-260) really made me curious about the context of the publications you reviewed. It would be very helpful to provide insights on context, particularly the country in which the studies were situated, as you allude to the fact that cultural variables may impact factors such as expectations, values, priorities, etc. of students, which may yield different implications for SX. 

You assert that "For this reason, we recommend that SX evaluation should be done at two levels, the first one focused on the nature of the touchpoints, and the second focused on the holistic analysis of the student's journey." (lines 639-641) - inclusion of a concrete example of each of these proposed levels would be helpful. 

This was a very well-executed and reported literature review. I believe it would make a valuable publication, especially if the noted issues were addressed. I am not certain it aligns well with this journal or the focus of this special issue - I will leave this to the editors to determine. If it is determined that it is not an appropriate fit, I encourage you to persist in your attempt to publish!

Author Response

Response to Reviewer 2 Comments

Point 1: It may be more accurate to revise your research questions to align with the criteria utilized to identify studies for inclusion in the review (i.e., specifically, the focus on undergraduate SX, as your results may not be generalizable to graduate-level SX).

Response 1: We appreciate your suggestion; we reviewed the research questions. Moreover, the updated version of the manuscripts systematically indicates its focus on undergraduate students in several parts of the document: abstract, introduction, research method, and conclusions.

Point 2: It would also be a helpful inclusion to briefly explain why you chose to exclude pedagogical articles (which seemingly have the potential to align with RQ 1&2), particularly given the assertion that "Since the concept's emergence, it has been referred to as a strictly pedagogical field. " (line 200). It would also be helpful to operationally define "highly technical applied cases" and explain why they were excluded. There is no problem with your decision to exclude what you did - it would just be helpful to briefly explain your rationale for such choices.

Response 2: We appreciate your comment. The updated manuscript includes an explanation in section 3.3. We did not actually exclude pedagogical articles; we only excluded articles narrowly focused on only pedagogical aspects. In our view, SX should be approached from a holistic perspective.

Point 3: You note that "Given the relevance of some articles referenced by those selected, it has been decided to include them despite not meeting the search criteria expressed in this review" (lines 144, 145). It would be helpful to provide examples or describe the criteria that made such articles relevant enough to be included despite not meeting the described criteria.

Response 3: We appreciate your suggestion. Section 6 now explains why additional references are relevant, providing justifications for their inclusion in our study.

Point 4: Your statement "We point out, like other authors, that the national and transnational discourses, government and institutional policies are factors that influence consumer orientation among students" (lines, 258-260) really made me curious about the context of the publications you reviewed. It would be very helpful to provide insights on context, particularly the country in which the studies were situated, as you allude to the fact that cultural variables may impact factors such as expectations, values, priorities, etc. of students, which may yield different implications for SX.

Response 4: Thank you for pointing out the missing context. The updated manuscript indicates that the study [57] refers to Chinese and Danish exchange students.

Point 5: You assert that "For this reason, we recommend that SX evaluation should be done at two levels, the first one focused on the nature of the touchpoints, and the second focused on the holistic analysis of the student's journey." (lines 639-641) - inclusion of a concrete example of each of these proposed levels would be helpful.

Response 5: We appreciate your suggestion. Section 5.3 now indicates concrete examples for both levels of SX evaluation.

Point 6: This was a very well-executed and reported literature review. I believe it would make a valuable publication, especially if the noted issues were addressed. I am not certain it aligns well with this journal or the focus of this special issue - I will leave this to the editors to determine. If it is determined that it is not an appropriate fit, I encourage you to persist in your attempt to publish!

Response 6: We really appreciate your very kind opinion. We hope that the updated manuscript attends all highlighted issues, and will be published.

This manuscript is a resubmission of an earlier submission. The following is a list of the peer review reports and author responses from that submission.